# Depressive Disorders and Sleeping Disturbances—Surveys Study of 923 Participants on the Pol’and’Rock Festival, Kostrzyn, Poland 2019

**DOI:** 10.3390/ijerph17218092

**Published:** 2020-11-03

**Authors:** Justyna Kałduńska, Karolina Skonieczna-Żydecka, Karina Ryterska, Joanna Palma, Wojciech Żwierełło, Dominika Maciejewska-Markiewicz, Maja Czerwińska-Rogowska, Anna Wolska, Zofia Stachowska, Robert Budawski, Honorata Mruk, Damian Matyniak, Magdalena Popik, Katarzyna Łoniewska, Marta Czmielnik, Aleksandra Fryda, Michal Wronski, Ewa Stachowska

**Affiliations:** 1Department of Human Nutrition and Metabolomics, Pomeranian Medical University in Szczecin, 71-460 Szczecin, Poland; justynakaldunska@wp.pl (J.K.); karzyd@pum.edu.pl (K.S.-Ż.); ryterska.karina@gmail.com (K.R.); palma.01.01@gmail.com (J.P.); domi.maciejka@wp.pl (D.M.-M.); majaczerwinska89@gmail.com (M.C.-R.); zbizcz@gmail.com (A.W.); ewast@pum.edu.pl (Z.S.); robertb93@o2.pl (R.B.); honorata.m@smsnet.pl (H.M.); damian-matyniak@wp.pl (D.M.); magdalena@popik.eu (M.P.); kloniewska@st.swps.edu.pl (K.Ł.); mczmielnik@gmail.com (M.C.); oiub@wp.pl (A.F.); 2Department of Medical Chemistry, Pomeranian Medical University in Szczecin, 70-111 Szczecin, Poland; chemia@pum.edu.pl; 3Department of Psychiatry, Pomeranian Medical University, 71-460 Szczecin, Poland; mwronski@pum.edu.pl

**Keywords:** Athens Insomnia Scale, Beck Inventory II Scale, cross-sectional study, depression, insomnia

## Abstract

Depressive disorders are common among young people and can decrease social competences and thus the quality of life. There is a relationship between the occurrence of depressive disorders and insomnia. The aim of the study was to determine the prevalence of insomnia and depressive behavior and assess the relationship between these among participants of the Pol’and’Rock Festival, Kostrzyn, Poland 2019. The study used the Athens Insomnia Scale (AIS) and the Beck Inventory II Scale (BDI-II). The study group consisted of 923 people, with the majority of women (*n* = 500; 54.2%). A total of 297 persons (32.2%) reported varying severity of depressive symptoms. Insomnia was observed in 261 (28.28%) respondents. Sleeping disturbances were observed more frequently in females. Persons with insomnia had a significantly higher BDI-II score. A strong positive correlation (r = 0.65) between the number of points obtained on the Beck and AIS scales was observed. Insomnia and depressive behavior are prevalent in the Polish population. Due to long-term social and economic consequences, special attention should be paid to the prevention, early detection and treatment of both disorders.

## 1. Introduction

Depression is a common mental disorder with clinical symptoms ranging from low mood to severe phenotype, including suicidal behavior [1]. The most common symptoms include sadness, guilt, loss of interest, inability to feel pleasure, fatigue, problems with sleep and difficulty concentrating [2]. It is estimated that over 268 million people worldwide suffer from depressive mood, of whom almost 168 million suffer from major depressive disorder (MDD) [3].

Depressive disorders are common among adolescents and may negatively affect social and family relationships [4]. The etiopathogenesis of depression is complex. It is assumed that the depressive phenotype development may be influenced by genetic, neuronal and environmental factors [5]. The neuronal predispositions include, e.g., changes in the structure of the brain (gray matter reduction) and disturbances in the functioning of the brain areas responsible for the regulation of emotional behavior [6]. The treatment is demanding; half of the patients respond inadequately to treatment, and while among the rest, the condition worsens. Additionally, in 40% of those who healed, the disease returns [7,8]. Chronic depressive disorders can cause weight loss, agitation or movement delay, difficulties to concentrate, sadness and sleep disturbances [9]. Finally, up to 80% of depressive patients complain of insomnia [10]. To add, the depressive mood has been found to significantly influence the development of somatic diseases [11].

Insomnia is defined as difficulties initiating sleep (DIS)/difficulties maintaining sleep (DMS) and non-restorative sleep (NRS) manifested by impaired daytime functioning for at least four weeks [12]. According to the International Statistical Classification of Diseases and Related Health Problems ICD-10 classification (International Statistical Classification of Diseases and Related Health Problems), insomnia is diagnosed if symptoms are present for at least one month and if other sleep disorders, the use of psychoactive substances and the coexistence of other somatic and psychiatric disorders have been excluded [13]. It is assumed that women experience insomnia more often than men [14], and the incidence of insomnia increases with age [10]. Insomnia significantly affects the deterioration of physical and mental health, leading to anxiety, more frequent use of stimulants, including alcohol and drugs [15]. Epidemiological studies indicate that up to 35% of the population experiences symptoms of insomnia, and sleep disorders are one of the most common complaints reported to primary care physicians [16].

There is also a correlation between the severity of depressive symptoms and insomnia [17]. It is a two-way phenomenon in which the presence of one disorder is a risk factor for the other [18]. It has been pointed out that insomnia has a significant impact on the occurrence of depressive mood, the severity of their symptoms, and the recurrence of depressive episodes [19]. Meta-analyses of epidemiological studies have shown that insomnia doubles the risk of depressive disorders in the following years [20]. Among people with an ongoing episode of major depression, the incidence is as high as 80% [21].

The aim of this study was to assess the frequency of sleep disorders and depressive symptoms among the Polish population and to analyze the relationship between the abovementioned symptoms. We decided to include persons around the age of 30 years predominantly, as the literature underlines a high incidence of depressive symptoms and insomnia in this age population.

## 2. Materials and Methods

### 2.1. Sampling and Data Collection

The survey was conducted in Kostrzyn, Poland, during the 25th Pol’and’Rock Festival (1–3 August 2019). The number of collected questionnaires each day was as follows: 1 August—352 questionnaires, 2 August—343 questionnaires and 3 August—228 questionnaires.

Festival participants were mainly young adults who came from all over the country; thus, the study may be considered as nationally representative. The data were collected by academic scientists and students from Pomeranian Medical University. The questionnaire was installed as an application on smartphones and tablets. Randomly selected participants located on the festival site and the associated campsites were asked to complete a questionnaire on their sleeping habits and mental well-being. Each of the respondents received a printed version of the questionnaire and answered the questions to the interviewer, who marked them to the application. The Medical Ethical Committee of the Pomeranian Medical University in Szczecin declared that as the questionnaire was anonymous, the respondents would not be contacted again, and their answers would not be stored in foreign internet servers. Written informed consent was not obligatory, but oral consent to take part in the study was obtained. Each study responder volunteered to participate in the experiment and personally pointed it out to the electronic device used in the study. The inclusion criteria were age above 18 years and being a Polish native speaker. Additionally, age, gender, body mass and height of participants were obtained.

### 2.2. Questionnaire

The Athens Insomnia Scale (AIS) and the Beck Depression Inventory (BDI-II) questionnaire were used for the study. Moreover, respondents answered the questions regarding sex, age, height and weight, as well as the educational stage.

BDI-II is a questionnaire that enables the assessment of the severity of depressive symptoms in adults and adolescents [22]. It consists of 21 issues, including 4 statements relating to the mental well-being of the last two weeks. The responses were scored on a 0–3 scale, increasing according to the severity of the symptoms [23]. The questions concerned 4 ways to support well-being: agitation, feelings of worthlessness, difficulty concentrating and loss of energy [24]. An interpretation of the questionnaire results is presented in Table 1.

AIS is a questionnaire for insomnia assessment based on the ICD-10 criteria. The test contains 8 questions relating to sleep habits. The 0–3 score scale corresponds to the severity of symptoms that occur at least 3 times a week in the last month [26]. A total score of ≥6 points was considered as a value that allowed inferring the presence of insomnia [27].

### 2.3. Statistical Analysis

Descriptive statistics were calculated to describe the demographic characteristics of the participants. The Shapiro–Wilk test was used to determine the normality of the continuous variables’ distribution, and nonparametric analyses were used as appropriate. Analyses were done using MedCalc software (version 19.4.3., Ostend, Belgium).

## 3. Results

### 3.1. Study Group

The study group consisted of 923 people, with the majority of women (*n* = 500; 54.2%). The characteristics of the group are presented in Table 2. The respondents differed in terms of education. The largest number of study participants—456 people (49.4%) declared higher education, 411 people (44.5%)—secondary education, 36 people (3.9%)—vocational education and 15 people (1.6%) declared no level of education.

### 3.2. Prevalence of Depressive Symptoms

Using the Beck Depression Inventory questionnaire (BDI-II), a group of 32.2% of the respondents reported depressive symptoms of varying severity (Table 3). The intensity of disturbances was not statistically associated with bodyweight or age (*p* < 0.05, Figure 1).

### 3.3. Presence of Insomnia

The analysis of the AIS score showed that a total of 261 people (28.3%) had symptoms of insomnia. The occurrence of insomnia was not statistically significantly associated with age (*p* = 0.5564) and body weight (*p* = 0.1237). There was also no statistically significant correlation between education and the occurrence of insomnia (*p* = 0.5984). On the other hand, statistically significant differences were observed between the occurrence of insomnia and gender (*p* < 0.0001). Sleep problems in a group of women were reported much more often.

Insomnia was confirmed in 171 women (median 5, IQR 2 to 9) and in 90 men (median 4, IQR 2 to 9). A detailed distribution is presented in Figure 2. The relationship between the respondents’ gender and the occurrence of sleep and falling asleep disorders, determined by the AIS questionnaire, was statistically significant (*p* = 0.000086).

### 3.4. Relationship between Insomnia and Depressive Symptoms

The respondents who showed symptoms of insomnia had a significantly higher score on the BDI-II scale (median 17, IQR 10 to 23; (*p* < 0.0001)), indicating mild depressive symptoms compared to those without sleep problems according to the AIS scale. The results are shown in Table 4.

Among people who did not report problems with sleep, most of the respondents did not have depressive symptoms either. On the other hand, in the group of respondents with insomnia, mild depressive symptoms occurred. A detailed distribution is presented in Table 4. Figure 3 shows the percentage distribution of the severity of depressive symptoms in the group of people with and without insomnia, and Figure 4 presents a detailed distribution of the number of points in the AIS scale in groups with various degrees of depressive symptoms.

The correlation between the number of points obtained in both questionnaires and the body weight and age of the respondents was also analyzed. A strong positive correlation was observed between BDI-II and AIS scores. Correlations are presented in Table 5.

Spearman’s rank correlation analysis was also carried out using the number of points in the AIS scale and body weight and age of the study participants. Negative correlations were found, but they were not statistically significant (Table 6).

## 4. Discussion

The present cross-sectional study was conducted to determine the prevalence of depressive disorders and insomnia in the sample of Polish adults and to search for dependencies in the occurrence of these phenotypes. The incidence of sleep disorders was 28.3%, and 32.2% of the respondents were found to suffer from depressive behavior of varying severity.

The prevalence of depressive symptoms in society is extensively analyzed in the literature. According to World Health Organization (WHO, Geneva, Switzerland) data, the number of people diagnosed with depression in the world is still growing. It has been estimated that over 264 million people currently suffer from depression all over the world [28]. The average prevalence rate of this entity among adult residents of European Union countries is over 11% [29]. In a study by Dróżdż et al., a total of 41% of respondents in the Polish population obtained a score of 12 points or more on the BDI-II scale and was clinically confirmed in 23.3% of the respondents [30]. In our study, depressive symptoms were reported in 32.2% of respondents, with the largest number of respondents showing its mild intensity.

In our study, the majority of the respondents (49.4%) had higher education. The relationship between the occurrence of anxiety and depression and the level of education is a subject of interest to many researchers [31,32,33]. A meta-analysis conducted by Lorant et al. points out that people with lower education have an increased risk of MDD compared to persons with higher education [34]. The hypotheses assume that economic status, social structure or cultural entitlement are commonly reduced among people with a low level of education [35]. Therefore, one should carefully pay attention to the relationship between the level of education and the number of points obtained in the BDI-II scale.

Interestingly, in this study, the BDI-II score was inversely correlated with the age of the respondents. This result, however, has been confirmed in the literature [30,36], which shows that in the population of young adults (including university students), depressive mood is a common disorder that significantly affects cognitive functions, academic performance, relationships with peers, self-worth, as well as mortality. It turned out that as many as 20–30% of students, despite experiencing significant depressive symptoms, do not use professional psychiatric help [37].

In our study, a total of 28.3% of respondents reported problems with their sleep, as evaluated by the AIS questionnaire. Epidemiological studies analyzing the frequency of sleep disorders show inconclusive results. Ohayon observed that the prevalence of insomnia ranges from 4.4% to 48%, depending on the research tool used [10]. A study of subjective insomnia in the Polish population, conducted by Nowicki et al. in a group of 2413 Poles, indicated that over half of the respondents (50.5%) reported sleep problems. Our results are similar to those obtained by Kiejna et al., who, in a study of 47.924 people over 15 years of age, observed that the incidence of insomnia in this population was 23.7% [38].

In this study, we also observed that the percentage of women struggling with insomnia was significantly higher (32.4%) than men (21.2%). This relationship was also confirmed by other researchers [39,40,41,42]. Furthermore, a meta-analysis published by Zhang and colleagues confirmed this phenomenon [43]. However, the link is not fully understood. Lindberg et al., studying the relationship between the gender of people with insomnia and their psychological status, observed that women more often report feelings of anxiety [44], which may deteriorate the quality of sleep [45,46]. It is also assumed that the increased frequency of insomnia in women compared to men is associated with a different hormonal balance; however, this mechanism has not been fully elucidated [47]. In our study, we did not observe any statistically significant correlation between insomnia and the age and weight of our respondents, which is opposite to Kiejna et al. results where the occurrence of insomnia was positively correlated with the age of the patients [39]. It should be highlighted that the population we examined was not representative (it is a weakness of our study)—the median age of responders in our study was 25 years. Meanwhile, sleep disorders tend to affect the elderly population, which was clearly indicated by Mazzotti and colleagues. They surveyed more than 16,600 participants aged 65 years (or older) and confirmed the frequent sleep disorders in this age group. What is important, in the present study, sleep complaints were associated with the presence of depression, mild cognitive impairment and a high number of comorbidities [48].

However, sleep disorders among young people can have a completely different cause, as shown in several meta-analyses. Results published by Short et al. proved that insufficient sleep is associated with 1.43 times greater odds of risky behaviors taken by adolescents, such as enhanced alcohol and drug use, increased violent behavior, including sexual risk, transport risk and others [49]. Another study on adolescents conducted by Hedin et al. showed that insomnia was associated with male gender, poor financial situation of a family and self-reported health, low physical activity and higher use of alcohol and/or cigarettes per month [50]. On the other hand, all these behaviors, as well as computer use (and evening light), may be related to sleep latency at this age [51].

Additionally, we noted that depressive symptoms were significantly more common among people who suffered from insomnia. Among the respondents with insomnia, as many as 68.97% indicated the presence of depressive symptoms of varying severity. The relationship between and insomnia has been widely reported in scientific reports. It is indicated that up to 90% of depressed patients complain of a significant reduction in sleep quality [52]. The inverse relationship is also known, in which insomnia is a predictor of depressive symptoms [53]. The relationship between these disorders is seen in the functioning of the brain stem and the thalamus nucleus, which are involved in the mechanisms modulating affective arousal and are involved in the pathophysiology of both sleep disorders and depressive disorders [54]. There are also other hypotheses of the occurrence of insomnia, which are partly related to the mechanisms regulating sleep [55], which include, among others, the hypothesis of monoamine deficiency [56] (dopamine, norepinephrine, adrenaline and serotonin) [57], the hypothesis of the corticosteroid receptor [58] or the hypothesis of circadian rhythm disturbances [59].

Furthermore, in a meta-analysis published by Becker and colleagues, it was shown that a strong relationship between sleep deprivation and depression exists [60]. What is interesting, in adults, both sleep duration as well deprivation may be risk factors for hypertension with stronger associations for women compared to men [61].

It seems that sleep disorders are associated with an increase in body inflammation, expressed by an increase in C-Reactive Protein (CRP) and Interleukin-6 (Il-6) concentration [62]. The reason for this may be the fact that poor sleep quality affects the activity of the hypothalamic-pituitary-adrenal (HPA) axis and the sympathetic nervous system (SNS), which turns in to increase the expression of inflammatory genes [63,64].

## 5. Conclusions

The results of our survey showed that both insomnia and depressive symptoms of varying severity are common disorders in young adults in Poland. Due to long-term social and economic consequences, special attention should be paid to the prevention, early detection and treatment of both disorders, especially in the coronavirus disease 2019 (COVID-19) era, where the pandemic situation intensifies both sleep disturbance and depressive behavior.

## Figures and Tables

**Figure 1 ijerph-17-08092-f001:**
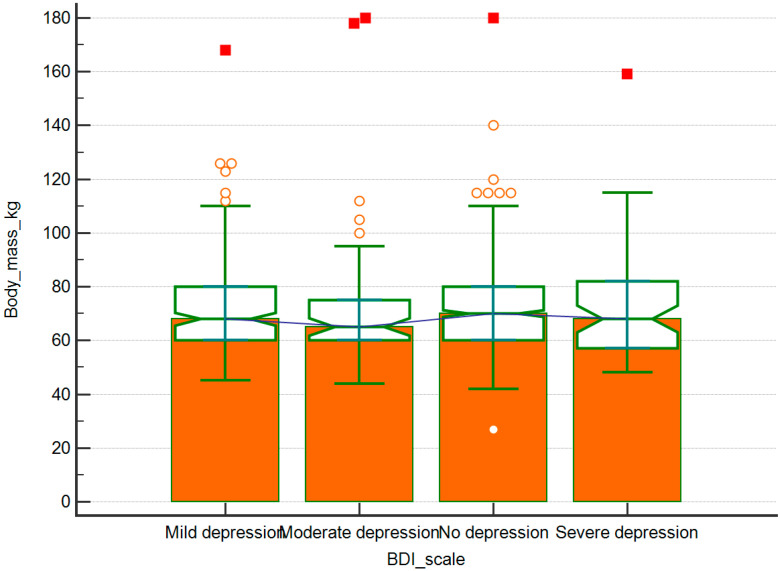
Bodyweight distribution in respondents with various degrees of depressive symptoms according to the BDI-II questionnaire.

**Figure 2 ijerph-17-08092-f002:**
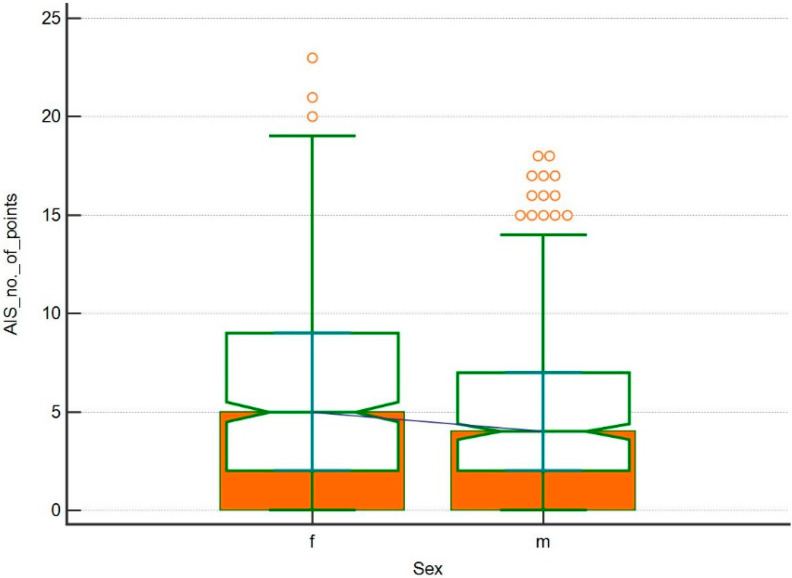
Number of points received by respondents in the Athens Insomnia Scale (AIS) questionnaire in relation to gender.

**Figure 3 ijerph-17-08092-f003:**
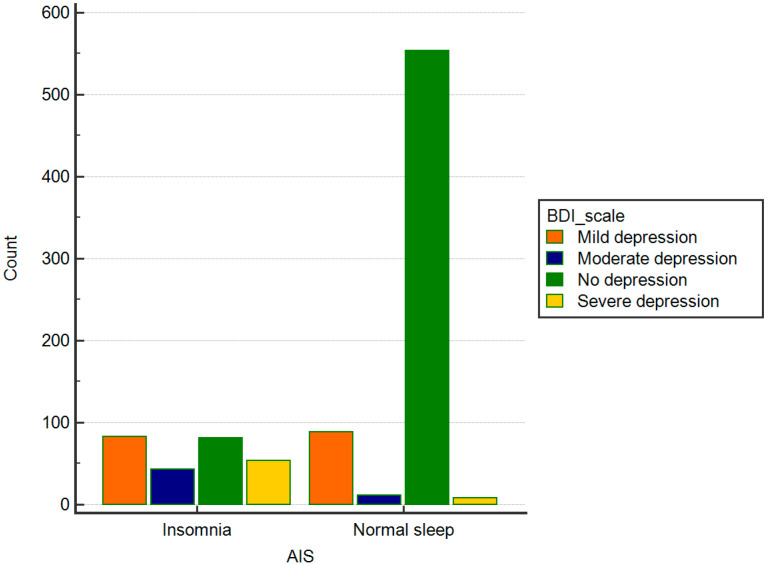
Percentage distribution of the severity of depressive symptoms according to the BDI-II classification in groups with and without insomnia according to the AIS scale.

**Figure 4 ijerph-17-08092-f004:**
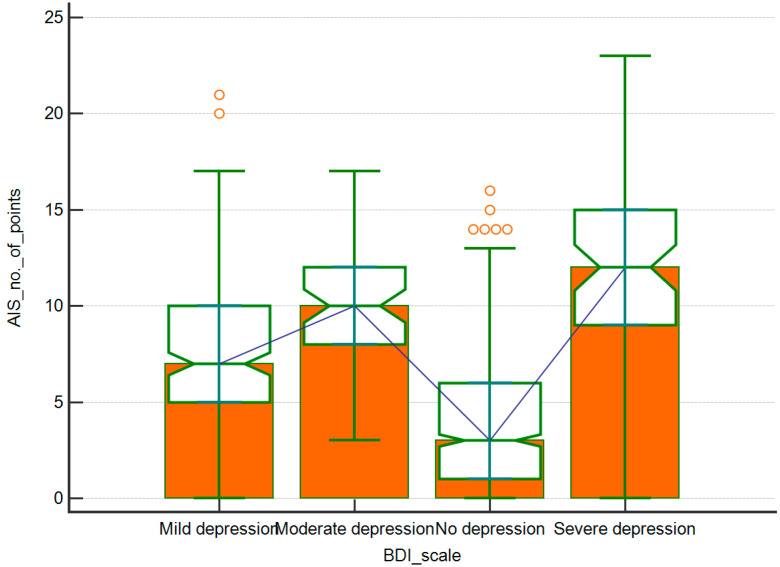
The number of points in the AIS scale obtained by respondents representing various levels of severity of depressive symptoms.

**Table 1 ijerph-17-08092-t001:** Interpretation of the Beck Inventory (BDI-II) questionnaire results [25].

Number of Points	Degree of Depressive Symptoms
0–13	Minimal range
14–19	Mild
20–28	Moderate
29–63	Severe

**Table 2 ijerph-17-08092-t002:** Characteristics of the study group.

Parameter	Median	IQR
Body mass (kg)	69	60.0 to 80.0
Age (years)	25	22.0 to 29.0
Height (cm)	172	165.0 to 180.0
AIS questionnaire score	4.0	2.0 to 8.0
Beck questionnaire score	7.0	3.0 to 14.0

**Table 3 ijerph-17-08092-t003:** Depressive symptoms among the study group.

Degree of Depressive Phenotype	N (%)
Minimal range	635 (68.8%)
Mild	172 (18.6%)
Moderate	54 (5.9%)
Severe	62 (6.7%)

**Table 4 ijerph-17-08092-t004:** Comparison of the severity of depressive symptoms in terms of the occurrence of insomnia using the chi-squared test (*p* < 0.0001). Numbers of subjects are presented.

**Variable**	**Minimal Depressive Symptoms**	**Mild Depressive Symptoms**	**Moderate Depressive Symptoms**	**Severe Depressive Symptoms**	***p***
Insomnia	81	83	43	54	
Normal sleep	554	89	11	8	<0.0001
total	635 68.8%	172 18.6%	54 5.9%	62 6.7%	

**Table 5 ijerph-17-08092-t005:** Correlation between BDI-II and AIS scores and body weight and age of the respondents by means of Spearman’s rank method.

**Number of BDI-II Points vs.:**	**Correlation Coefficient**	***p***
Number of AIS points	0.65	<0.0001
Age	−0.124	0.0002
Body weight	−0.061	0.0644

**Table 6 ijerph-17-08092-t006:** Correlation between Beck Depression Inventory questionnaire (BDI-II) and AIS scores and body weight and age of the respondents by means of Spearman’s rank method.

**Number of AIS Points vs.:**	**Correlation**	***p***
Number of BDI-II points	0.65	<0.0001
Age	−0.041	0.2169
Body weight	−0.044	0.1817

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
