# Peer review of "Depressive Disorders and Sleeping Disturbances—Surveys Study of 923 Participants on the Pol’and’Rock Festival, Kostrzyn, Poland 2019"

_ijerph, 2020, doi:10.3390/ijerph17218092_

Round 1
Reviewer 1 Report
This is a good work because you have done a very good field surveys across the entire territory and statistical analyses. Otherwise, I have a lot of recommendations to increase the quality of your paper. Be careful with the writing and mistakes.
The keyword “depressive disorder” is repeated in the article title. In order to increase the visibility of your paper I recommend changing the keyword. If you change it by a different keyword, you will increase the probability that your paper could be found by future readers when they look for your paper in some databases like Scopus for example. If you repeat the same words in the article title and in keywords, less people could find your work. So, you must think about the visibility of your research.
Please, put the keywords in alphabetical order. The journal publishes the keywords in this way. Follow the rules of the journal.
I have found a very recent paper which title is: “Insomnia in Relation to Academic Performance, Self-Reported Health, Physical Activity, and Substance Use Among Adolescents” so I think that is very interesting to read or use this paper in order to increase the quality of your article.
In line 38 write in capitals the letters used for the acronym: so, you should write “Major Depressive Disorder” just before “(MDD)”, this is very useful for a fluent reading.
In line 51 you should make the same corrections as the former one. So, you must write “Difficulties to Initiate Sleep” just before “(DIS)” and “Difficulties to Maintain Sleep” just before (“DMS”).
The same in line 52: “Non-Restorative Sleep” just before “(NRS)”. Please, look for the same mistake in all your paper to correct this tiny mistake.
You must be more explicit in the date the people answered the questionnaire. You must write in the “Sampling and data collection” the accurate date they answered after or during the Festival. This is not very clear. This is very important because the answers may be different depending on the date you asked to the people. Try to write as much clear as you can.
In line 169 you have written “WHO”, but I think that in this case you must write “(World Health Organization)” just after the word “WHO”. This make the reading experience easier for a potential non-expert reader, and this increase the visibility of your work.
In line 231 you must write “C-Reactive Protein” just before “(CRP)”.
In line 232 you must write “Hypothalamic-Pituitary-Adrenal” just before “(HPA)”.
And finally, in line 233 you must write “Sympathetic Nervous System” just before “(SNS)”.
Even in the conclusions just after COVID you must write into brackets “(COronaVIrus Desease)”. This is very important because some people confuse the virus SARS-CoV-2 with its desease which is COVID-19. In the conclusions instead of “COVID” I would change it by “COVID-19”.
Author Response
October 7th, 2020
Dear Editors-in-Chief,
I take the liberty to thank you and the reviewers for insightful and careful evaluation of our article entitled “Depressive disorders and sleeping disturbances - surveys study of 923 participants on the Pol’and’rock Festival, Kostrzyn, Poland 2019” (ijerph-940642) and for allowing us to resubmit a revised manuscript.
The comments helped us to improve the quality of the manuscript. We considered all comments and recommendations and responded to Reviewers’ questions. The correction throughout the manuscript were done using track changes mode.
Our responses to the reviews are attached below – marked in red.
Thank you for your consideration. I look forward to hearing from you.
Sincerely,
Ewa Stachowska
RESPONSES TO THE REVIEWERS’ COMMENTS:
We wish to thank you all for your constructive comments in this review. Your comments provided valuable insights to refine our manuscript. In this document, we try to address the issues raised as best as possible.
Reviewer #1:
This is a good work because you have done a very good field surveys across the entire territory and statistical analyses. Otherwise, I have a lot of recommendations to increase the quality of your paper. Be careful with the writing and mistakes.
Thank you very much for being such a kind and helpful review. We have tried our best to apply to all valuable comments.
The keyword “depressive disorder” is repeated in the article title. In order to increase the visibility of your paper I recommend changing the keyword. If you change it by a different keyword, you will increase the probability that your paper could be found by future readers when they look for your paper in some databases like Scopus for example. If you repeat the same words in the article title and in keywords, less people could find your work. So, you must think about the visibility of your research.
We have changed and added the keywords.
Please, put the keywords in alphabetical order. The journal publishes the keywords in this way. Follow the rules of the journal.
We have changed order of the keywords.
I have found a very recent paper which title is: “Insomnia in Relation to Academic Performance, Self-Reported Health, Physical Activity, and Substance Use Among Adolescents” so I think that is very interesting to read or use this paper in order to increase the quality of your article.
Thank you for pointing to the above-mentioned work. The results of this research were used by us to develop the Discussion section.
In line 38 write in capitals the letters used for the acronym: so, you should write “Major Depressive Disorder” just before “(MDD)”, this is very useful for a fluent reading.
We have applied corrections suggested by the reviewer.
In line 51 you should make the same corrections as the former one. So, you must write “Difficulties to Initiate Sleep” just before “(DIS)” and “Difficulties to Maintain Sleep” just before (“DMS”).
We have applied corrections suggested by the reviewer.
The same in line 52: “Non-Restorative Sleep” just before “(NRS)”. Please, look for the same mistake in all your paper to correct this tiny mistake.
We have applied corrections suggested by the reviewer.
You must be more explicit in the date the people answered the questionnaire. You must write in the “Sampling and data collection” the accurate date they answered after or during the Festival. This is not very clear. This is very important because the answers may be different depending on the date you asked to the people. Try to write as much clear as you can.
The reviewer rightly pointed out that the date of collecting the questionnaire may affect the responses of the respondents. The questionnaires were collected during the festival (1-3 August 2019). The exact number of questionnaires completed by the respondents each day have been included in the body of the article.
In line 169 you have written “WHO”, but I think that in this case you must write “(World Health Organization)” just after the word “WHO”. This make the reading experience easier for a potential non-expert reader, and this increase the visibility of your work.
We have applied corrections suggested by the reviewer.
In line 231 you must write “C-Reactive Protein” just before “(CRP)”.
We have applied corrections suggested by the reviewer.
In line 232 you must write “Hypothalamic-Pituitary-Adrenal” just before “(HPA)”.
We have applied corrections suggested by the reviewer.
And finally, in line 233 you must write “Sympathetic Nervous System” just before “(SNS)”.
We have applied corrections suggested by the reviewer.
Even in the conclusions just after COVID you must write into brackets “(COronaVIrus Desease)”. This is very important because some people confuse the virus SARS-CoV-2 with its desease which is COVID-19. In the conclusions instead of “COVID” I would change it by “COVID-19”.
We have applied corrections suggested by the reviewer.
Reviewer 2 Report
I think that the article can be published after significantly improvements.
Considering that the sample surveyed is somewhat particular, the background of the study must be changed toward this specificity. In the same time, it is necessary a changing of the presentation the results of the statistical analysis.
I think that must be emphasized the characteristics between the adolescents and young adults and also the high percentage of the persons with high education must pay more attention. Detailed comment abouts this can be made in the Discussion section.
I think a careful adjustment in all part of the article can made it appropriate for publication.
Author Response
October 7th, 2020
Dear Editors-in-Chief,
I take the liberty to thank you and the reviewers for insightful and careful evaluation of our article entitled “Depressive disorders and sleeping disturbances - surveys study of 923 participants on the Pol’and’rock Festival, Kostrzyn, Poland 2019” (ijerph-940642) and for allowing us to resubmit a revised manuscript.
The comments helped us to improve the quality of the manuscript. We considered all comments and recommendations and responded to Reviewers’ questions. The correction throughout the manuscript were done using track changes mode.
Our responses to the reviews are attached below – marked in red.
Thank you for your consideration. I look forward to hearing from you.
Sincerely,
Ewa Stachowska
RESPONSES TO THE REVIEWERS’ COMMENTS:
We wish to thank you all for your constructive comments in this review. Your comments provided valuable insights to refine our manuscript. In this document, we try to address the issues raised as best as possible.
Reviewer #2:
I think that the article can be published after significantly improvements.
Considering that the sample surveyed is somewhat particular, the background of the study must be changed toward this specificity.
We agree with the reviever that the studied group is of a “specific” type. The explanation for the selection of such study group has been included into Introduction section.
In the same time, it is necessary a changing of the presentation the results of the statistical analysis.
Thank you for your opinion. However, we are concerned that a different presentation of our research results could adversely affect their interpretation. At the same time, we are open to any suggestions on how to present our results so that they are as much easy to follow by readers as possible.
I think that must be emphasized the characteristics between the adolescents and young adults and also the high percentage of the persons with high education must pay more attention. Detailed comment abouts this can be made in the Discussion section.
The reviewer rightly pointed out that there is a difference between adolescents and young adults. We had used this two words as an replacement but it was incorrect. We have changed that mistake. We have also paid more attention to relationship between high education level of our participants and the occurrence of depressive disorders. These considerations have been placed in Discussion section.
I think a careful adjustment in all part of the article can made it appropriate for publication.
Once again, we thank you for the time you put in reviewing our paper and look forward to meeting your expectations. Since your inputs have been precious, in the eventuality of a publication, we would like to acknowledge your contribution explicitly.
Reviewer 3 Report
This manuscript examines the correlation between insomnia symptoms and depressive symptoms in young adults attending a music festival, in a cross-sectional design. Given the large body of literature examining depressive symptoms and insomnia (including longitudinal predictive analyses, meta-analyses, and complex examinations of the bidirectional relationship between insomnia and depression), there does not seem to be a strong rationale for examining a correlation between these two questionnaires. It is also unclear what the significance of the music festival attendance is, other than convenience.Author Response
October 7th, 2020
Dear Editors-in-Chief,
I take the liberty to thank you and the reviewers for insightful and careful evaluation of our article entitled “Depressive disorders and sleeping disturbances - surveys study of 923 participants on the Pol’and’rock Festival, Kostrzyn, Poland 2019” (ijerph-940642) and for allowing us to resubmit a revised manuscript.
The comments helped us to improve the quality of the manuscript. We considered all comments and recommendations and responded to Reviewers’ questions. The correction throughout the manuscript were done using track changes mode.
Our responses to the reviews are attached below – marked in red.
Thank you for your consideration. I look forward to hearing from you.
Sincerely,
Ewa Stachowska
RESPONSES TO THE REVIEWERS’ COMMENTS:
We wish to thank you all for your constructive comments in this review. Your comments provided valuable insights to refine our manuscript. In this document, we try to address the issues raised as best as possible.
Reviewer #3:
This manuscript examines the correlation between insomnia symptoms and depressive symptoms in young adults attending a music festival, in a cross-sectional design.
Given the large body of literature examining depressive symptoms and insomnia (including longitudinal predictive analyses, meta-analyses, and complex examinations of the bidirectional relationship between insomnia and depression), there does not seem to be a strong rationale for examining a correlation between these two questionnaires.
Due to the presence of many scientific reports on the coexistence of insomnia and depression, in our study we wanted to check to what extent they are dependent on each other. The correlation analysis allowed us to conclude that depressive disorders are common among people with insomnia symptoms and the occurrence of these two diseases is related to each other.
It is also unclear what the significance of the music festival attendance is, other than convenience.
The reviewer questioned the choice of music festival as a place to conduct the study. We would like to emphasize that the Pol'and'Rock Festival (previously known as the Woodstock Festival) has been organized for 26 years. It is the largest music festival in Poland and one of the largest rock festivals in Europe. Hundreds of thousands of people from all over Poland take part in it every year. Therefore, its participants can be considered a representative sample for the entire country, as also described in our study. We have also included a mention of this fact in the Discussion section.
Round 2
Reviewer 2 Report
The changesets of the text improve significantly the accuracy of the method and the aims, specificity of this study. This are clearer put in evidence and so the value of the article is putted in evidence.
The explanation of the initials is welcome for the accuracy of the text.
As a result of this changes, I consider that the article can be published, after some minor spelling of the text.
Author Response
RESPONSES TO THE REVIEWERS’ COMMENTS:
We wish to thank you all for your constructive comments in this review. Your comments provided valuable insights to refine our manuscript. In this document, we try our best to explain the reported problems.
Reviewer #2:
The changesets of the text improve significantly the accuracy of the method and the aims, specificity of this study. This are clearer put in evidence and so the value of the article is putted in evidence.
The explanation of the initials is welcome for the accuracy of the text.
We are thankful for this comment. We have presented the explanation of the initials in the Author Contributions section.
As a result of this changes, I consider that the article can be published, after some minor spelling of the text.
We are thankful for your review. We also made a slight linguistic correction to remove typing errors. Thank you for any comments that have significantly improved the quality of our work.
We believe that our work clearly shows that the problem of depressive disorders and sleep disorders is a serious problem for young Poles. Our next step will be to conduct a study combining our two topics of interest, i.e. the analysis of the relationship between disorders of the digestive tract and disorders of the intestinal microflora, and the mental health disturbances.
In this study we also wanted to emphasize the importance of mental health monitoring, especially in the current pandemic of COVID-19.
We hope that our explanation will convince the Reviewer and allow us to publish our work.

Reviewer 3 Report
Unfortunately, my original concern regarding the lack of novelty or advancement of the field in the current manuscript still stands.
A Scopus search for "depression" or "depressive" AND "insomnia" returns 23,313 publications. This manuscript reports a correlation between two scales, highlighting an association that is well-documented, and has been examined extensively and much more thoroughly in previous work. This correlation does not add to or expand upon our current knowledge.
Author Response
We wish to thank you all for your constructive comments in this review. Your comments provided valuable insights to refine our manuscript. In this document, we try our best to explain the reported problems.
Reviewer #3:
Unfortunately, my original concern regarding the lack of novelty or advancement of the field in the current manuscript still stands.
A Scopus search for "depression" or "depressive" AND "insomnia" returns 23,313 publications. This manuscript reports a correlation between two scales, highlighting an association that is well-documented, and has been examined extensively and much more thoroughly in previous work. This correlation does not add to or expand upon our current knowledge.
Thank you for your opinion. We are fully aware that our research focuses on a topic that has long been thoroughly researched and described. The aim of our study was to investigate the occurrence of depressive disorders and sleep disorders in a specific age group, i.e. young polish adults. We have been conducting our research at the Pol'and'Rock festival (previously Woodstock Festival) for many years. So far, our research has focused on intestinal disorders, intestinal microbiota and also the gut- brain axis, which is proved by our work (Skonieczna- Żydecka et al., The Digestive Health among Participants of the Woodstock Rock Festival in Poland—A CrossSectional Survey.; Stachowska et al., Abdominal Pain and Disturbed Bowel Movements are Frequent among Young People. A Population Based Study in Young Participants of the Woodstock Rock Festival in Poland).
During these studies, we observed that most of our respondents, while pointed to gastrointestinal complaints, also mentioned depressed mood and problems with falling asleep. Therefore, we decided to check the frequency of these disorders among festival participants, as evidenced by the above research.
We believe that our work clearly shows that the problem of depressive disorders and sleep disorders is a serious problem for young Poles. Our next step will be to conduct a study combining our two topics of interest, i.e. the analysis of the relationship between disorders of the digestive tract and disorders of the intestinal microflora, and the mental health disturbances.
In this study we also wanted to emphasize the importance of mental health monitoring, especially in the current pandemic of COVID-19.
We hope that our explanation will convince the Reviewer and allow us to publish our work.
